# Studies on the Enantioselective Synthesis of *E*-Ethylidene-bearing Spiro[indolizidine-1,3′-oxindole] Alkaloids

**DOI:** 10.3390/molecules26020428

**Published:** 2021-01-15

**Authors:** Nihan Yayik, Maria Pérez, Elies Molins, Joan Bosch, Mercedes Amat

**Affiliations:** 1Laboratory of Organic Chemistry, Faculty of Pharmacy and Food Sciences, and Institute of Biomedicine (IBUB), University of Barcelona, 08028 Barcelona, Spain; nhnyyk93@gmail.com (N.Y.); joanbosch@ub.edu (J.B.); 2Department of Nutrition, Food Sciences and Gastronomy, Faculty of Pharmacy and Food Sciences, and Institute of Nutrition and Food Safety (INSA-UB), University of Barcelona, 08921 Santa Coloma de Gramanet, Spain; mariaperez@ub.edu; 3Institut de Ciència de Materials de Barcelona (ICMAB-CSIC), Campus UAB, 08193 Cerdanyola, Spain; elies.molins@icmab.es

**Keywords:** oxindoles, alkaloids, spiro compounds, ethylidene

## Abstract

A synthetic route for the enantioselective construction of the tetracyclic spiro[indolizidine-1,3′-oxindole] framework present in a large number of oxindole alkaloids, with a *cis* H-3/H-15 stereochemistry, a functionalized two-carbon substituent at C-15, and an *E*-ethylidene substituent at C-20, is reported. The key steps of the synthesis are the generation of the tetracyclic spirooxindole ring system by stereoselective spirocyclization from a tryptophanol-derived oxazolopiperidone lactam, the removal of the hydroxymethyl group, and the stereoselective introduction of the *E*-ethylidene substituent by acetylation at the α-position of the lactam carbonyl, followed by hydride reduction and elimination. Following this route, the 21-oxo derivative of the enantiomer of the alkaloid 7(*S*)-geissoschizol oxindole has been prepared.

## 1. Introduction

The spiro[pyrrolidine-3,3′-oxindole] ring system is a structural moiety found in a large number of natural products, pharmaceuticals, and biologically active compounds. This moiety is present in oxindole alkaloids, which constitute a large group of monoterpenoid alkaloids characterized by a spiro-fusion to a pyrrolidine ring at the 3-position of the indole core [1,2,3]. Oxindole alkaloids possess a variety of pharmacological properties [4,5,6,7,8,9,10,11,12] and have served as an inspiration for the development of new therapeutic agents [13]. One of the most important substructural classes of these alkaloids incorporates a tetracyclic spiro[indolizidine-1,3′-oxindole] framework, in which the pyrrolidine nucleus is embedded in an indolizidine ring. Structurally, these tetracyclic oxindole alkaloids differ in the functionalized substituent at C-15 and the two-carbon substituent at C-20 (usually ethyl, vinyl or *E*-ethylidene), as well as in the configuration at the C-3, C-7, C-15, and (in some cases) C-20 stereocenters (biogenetic numbering) [14], thus giving rise to a diversified array of relative stereochemical relationships. Some representative examples are depicted in Figure 1. 

The members of this group of alkaloids often occur in nature as pairs of C-7 epimers [15,16], interconvertible under acidic or basic conditions by retro-Mannich/Mannich reactions via a ring-opened intermediate [17,18,19,20]. Scheme 1 illustrates this equilibrium for geissoschizol oxindoles [16]. 

The appealing structure of these alkaloids and their significant biological activities have attracted considerable synthetic attention, resulting in a large number of total syntheses, both in the racemic series and in enantiopure form [21,22,23,24]. However, the access to *E*-ethylidene-bearing spiro[indolizidine-1,3′-oxindoles] with a *cis* H-3/H-15 stereochemistry has been little explored and to our knowledge no total syntheses of tetracyclic oxindole alkaloids with this substitution and stereochemical pattern have been reported so far. In fact, although the oxidative rearrangement of tetrahydro-β-carbolines is one of the most frequently used methods to assemble the spiro[pyrrolidine-3,3′-oxindole] system [18,19,21], its application to either *cis* or *trans E*-ethylidene-bearing *Corynanthe*-type indolo[2,3-*a*]quinolizidine amines (but not amides) leads only to C-7 epimeric mixtures of tetracyclic spirooxindoles with a *trans* H-3/H-15 stereochemistry due to an equilibration of the C-3 and C-7 stereocenters via a reversible Mannich reaction (Scheme 2) [25,26]. The severe A^1,3^-interaction between the C-15 side chain and the methyl group on the ethylidene moiety in the *cis* isomers, which is absent in the *trans* isomers, could account for the exclusive formation of the latter [26].

## 2. Results and Discussion

We present herein a procedure for the enantioselective construction of *cis* H-3/H-15 tetracyclic spiro[indolizidine-1,3′-oxindoles] bearing an *E*-ethylidene substituent at the C-20 position. The *cis* H-3/H-15 relationship was secured using the spirocyclization procedure we have recently reported for the direct generation of spirooxindoles from tryptophanol-derived oxazolopiperidone lactams [27], whereas the *E*-ethylidene substituent was stereoselectively installed, taking advantage of the piperidone carbonyl present in the resulting tetracyclic spirooxindole. Scheme 3 outlines the initial steps of the synthesis.

The required bicyclic lactam **2** was prepared, as previously reported [28], by cyclocondensation of prochiral aldehyde-diester **1** with (*S*)-tryptophanol in a process that involves the stereoselective desymmetrization of two enantiotopic acetate chains. A subsequent bromination with pyridinium perbromide, with careful control of the reaction time (10 s) and operating under strictly anhydrous conditions to minimize the formation of the corresponding oxindole, afforded 2-bromoindole **3** in an improved yield (92%). The latter underwent smooth spirocyclization on treatment with TFA to provide (71%) tetracyclic spirooxindole **4** as a single stereoisomer, whose absolute configuration was unambiguously confirmed by X-ray crystallographic analysis (Figure 2; see Appendix A).

Removal of the hydroxymethyl substituent, which plays a decisive role as an element of stereocontrol in the spirocyclization reaction, was initially accomplished in four steps: oxidation to carboxylic acid **6** via the corresponding aldehyde **5**, subsequent generation of selenoester **7**, and finally radical reductive decarbonylation (Scheme 4) [29,30]. Following this procedure, tetracyclic oxindole **8** was obtained in 45% overall yield from aldehyde **5**.

Alternatively, **8** was prepared in a more efficient manner by direct rhodium(I)-induced decarbonylation of aldehyde **5**. Thus, treatment of **5** with the rhodium(I) complex generated in situ from the chloro(1,5-cyclooctadiene)rhodium(I) dimer {[RhCl(cod)]_2_} and the bidentate phosphine ligand 1,3-bis(diphenylphosphino)propane (dppp) [31] satisfactorily provided oxindole **8** in 79% yield. The use of the rhodium(I) complex derived from carbonyl(chloro)bis(triphenylphosphine)rhodium(I) [RhCl(CO)(PPh_3_)_2_] and dppp [32] gave less satisfactory results (46% yield).

The preparation of **8** in four steps and 40% overall yield from tryptophanol-derived lactam **2** represents a notable improvement with respect to the previously reported eight-step route from the same starting lactam **2**, involving the generation of a spiroindoline and the final oxidation of an indoline [27,28], which gave **8** in 13% overall yield (Scheme 5).

Having secured an efficient and reliable sequence to access the tetracyclic scaffold **8**, we focused on the introduction of the C-20 two-carbon substituent. Chemoselective LiBH_4_ reduction of the ester group and subsequent successive protection of the resulting primary alcohol **9** with *tert*-butydimethylsilyl chloride (TBDMSCl) and the indole nitrogen with either methoxymethyl chloride (MOMCl) or di-*tert*-butyl dicarbonate (Boc_2_O) afforded the diprotected intermediates **12** and **13** (Scheme 6). Under the LiBH_4_ reduction conditions, only minor amounts (5%) of the corresponding indoline **10** were formed.

Initial attempts to directly introduce a 1-hydroxyethyl chain by an aldol-type reaction between **12** or **13** and acetaldehyde (LDA, −78 °C) were unsuccessful, and the corresponding alcohols were only formed in trace amounts. Satisfactorily, the *E*-ethylidene substituent was installed by acylation at the α-position of the lactam carbonyl of **12**, followed by hydride reduction and stereoselective elimination. Thus, acetylation of **12** with methyl acetate in the presence of LDA afforded ketone **14** as a single stereoisomer in 52% yield, with 29% of the starting material being recovered. Although a subsequent reduction with NaBH_4_ gave a nearly equimolecular mixture of epimeric alcohols **15a** and **15b**, they could be separated by column chromatography and independently converted to the *E*-ethylidene derivative **16** in excellent overall yield.

Alcohol **15a** was dehydrated by treatment with DCC and CuCl as a catalyst in refluxing toluene [33,34,35] to stereoselectively provide the *E*-ethylidene derivative **16** in 91% yield. This is a useful method for the *syn* elimination of β-hydroxycarbonyl compounds that proceeds via an isourea intermediate through a six-membered cyclic transition state [36]. In turn, alcohol **15b** was subjected to an *anti*-elimination sequence [34,35] by treatment of the corresponding mesylate with DBU to give the expected *E*-ethylidene derivative **16** (60% yield) along with minor amounts of its *Z* isomer (20% yield). The stereochemistry of both isomers was assigned from their ^1^H-NMR spectra, in which the olefinic proton of the *E* isomer appears at a lower field (δ 6.67) than that of the *Z* isomer (δ 5.94) due to the anisotropic effect of the lactam carbonyl group. A final treatment of **16** under smooth conditions brought about the simultaneous deprotection of the oxindole and alcohol moieties to provide the *cis* H-3/H-15, *E*-ethylidene-bearing spiro[indolizidine-1,3′-oxindole] **17**.

In summary, we have developed a procedure for the stereoselective construction of the spiro[indolizidine-1,3′-oxindole] framework characteristic of tetracyclic oxindole alkaloids, bearing a *cis* H-3/H-15 stereochemistry and incorporating a functionalized two-carbon substituent at C-15 and an *E*-ethylidene substituent at C-20. These studies could open synthetic routes to tetracyclic oxindole alkaloids featuring this substitution and stereochemical pattern, for instance, 7(*S*)-kopsirensine A or 7(*S*)-isositsirikine oxindole. In fact, compound **17** can be envisaged as the enantiomer of 21-oxo-7(*S*)-geissoschizol oxindole. The use of (*R*)-tryptophanol instead of the *S* enantiomer in the synthetic sequence outlined in Scheme 6 would provide access to the natural enantiomeric series.

## 3. Experimental Section

### 3.1. General Information

All air sensitive manipulations were carried out under a dry argon or nitrogen atmosphere. THF and CH_2_Cl_2_ were dried using a column solvent purification system. Analytical thin-layer chromatography was performed on SiO_2_ (Merck silica gel 60 F254), and the spots were located with 1% aqueous KMnO_4_. Chromatography refers to flash chromatography and was carried out on SiO_2_ (SDS silica gel 60 ACC, 35–75 mm, 230–240 mesh ASTM). NMR spectra were recorded at 400 (^1^H) and 100.6 (^13^C), and chemical shifts are reported in δ values downfield from TMS or relative to residual chloroform (7.26 ppm, 77.0 ppm) as an internal standard. Data are reported in the following manner: Chemical shift, multiplicity, coupling constant (*J*) in hertz (Hz), integrated intensity, and assignment (when possible). Assignments and stereochemical determinations are given only when they are derived from definitive two-dimensional NMR experiments (HSQC-COSY). IR spectra were performed in a spectrophotometer Nicolet Avantar 320 FT-IR, and only noteworthy IR absorptions (cm^−1^) are listed. High resolution mass spectra (HMRS) were performed by Centres Científics i Tecnològics de la Universitat de Barcelona.

### 3.2. Improved Preparation of Tetracyclic Oxindole ***4***

*(3S,7R,8aS)-3-[(2-Bromo-3-indolyl)methyl]-7-(methoxycarbonylmethyl)-5-oxo-2,3,6,7,8,8a-hexahydro-5H-oxazolo[3,2-a]pyridine* (**3**). A solution of pyridinium tribromide (PyHBr_3_, 4.20 g, 13.1 mmol) in anhydrous THF (57 mL) was added at 0 °C by a large transfer, under an argon atmosphere, to a stirred solution of lactam **2** [28] (3.21 g, 9.36 mmol) in anhydrous CH_2_Cl_2_ (57 mL). Saturated aqueous solutions of Na_2_S_2_O_3_ (19 mL) and NaHCO_3_ (10 mL) were added 10 s later. Distilled H_2_O (10 mL) was added, and the mixture was extracted with CH_2_Cl_2_. The combined organic extracts were dried over anhydrous MgSO_4_, filtered, and concentrated under reduced pressure. Flash chromatography (hexane to 7:3 hexane-EtOAc) of the resulting residue gave bromo derivative **3** (3.63 g, 92%) as a white foam: [α]^22^_D_ = −74.1 (*c* 1.4, CHCl_3_); IR (KBr): 3171 (NH), 1630, 1734 (CO) cm^−1^; ^1^H-NMR (400 MHz, CDCl_3_, COSY, g-HSQC): *δ* = 1.31–1.40 (qm, *J* = 11.6 Hz, 1H, H-8), 2.17 (dd, *J* = 17.6, 10.4 Hz, 1H, H-6), 2.32 (dm, *J* = 11.6 Hz, 1H, H-8), 2.37 (m, 1H, H-7), 2.40 (s, 2H, CH_2_CO_2_), 2.65 (dd, *J* = 17.6, 4.8 Hz, 1H, H-6), 2.69 (dd, *J* = 14.0, 10.4 Hz, 1H, CH_2_-Ind), 3.65–3.69 (m, 2H, CH_2_-Ind, H-2), 3.69 (s, 3H, CH_3_O), 4.08 (d, *J* = 8.8 Hz, 1H, H-2), 4.31 (ddd, *J* = 9.6, 6.8, 3.2 Hz, 1H, H-3), 4.70 (dd, *J* = 10.0, 3.2 Hz, 1H, H-8a), 7.07 (td, *J* = 8.0, 0.8 Hz, 1H, H_AR_), 7.13 (td, *J* = 8.0, 0.8 Hz, 1H, H_AR_), 7.28 (dd, *J* = 8.0, 0.8 Hz, 1H, H_AR_), 7.76 (dd, *J* = 8.0, 0.8 Hz, 1H, H_AR_), 9.28 (br. s, 1H, NH); ^13^C-NMR (100.6 MHz, CDCl_3_): *δ* = 26.4 (CH_2_-Ind), 27.2 (C-7), 34.4 (C-8), 37.6 (C-6), 40.1 (*C*H_2_CO), 51.8 (CH_3_O), 55.5 (C-3), 70.2 (C-2), 88.3 (C-8a), 108.8 (C-Br), 110.4 (CH_AR_), 112.0 (C_AR_), 118.7 (CH_AR_), 120.3 (CH_AR_), 122.5 (CH_AR_), 127.9 (C_AR_), 136.1 (C_AR_), 166.8 (CO), 171.8 (CO); HRMS (ESI) calcd for [C_19_H_22_BrN_2_O_4_ + Na^+^]: 433.0577, found: 433.0577.

*(1’R,3′S,7′R,8a’R)-3′-(Hydroxymethyl)-7′-(methoxycarbonylmethyl)-2,5′-dioxospiro[indoline-3,1′-indolizidine]* (**4**). TFA (3.15 mL, 40.9 mmol) was added at room temperature under an argon atmosphere to a solution of bromo derivative **3** (556 mg, 1.32 mmol) in anhydrous CH_2_Cl_2_ (25 mL). The resulting mixture was stirred at room temperature for 16 h and then concentrated under reduced pressure. Flash chromatography (silica previously washed with Et_3_N; 1:1 to 1:9 hexane-EtOAc) of the resulting residue gave spirooxindole **4** (337 mg, 71%) as a white foam: [α]^22^_D_ = + 46.1 (*c* 1.0, CHCl_3_); IR (KBr): 3401 (NH), 1617, 1724 (CO) cm^−1^; ^1^H-NMR (400 MHz, CDCl_3_, COSY, g-HSQC): *δ* = 0.69 (q, *J* = 12.4 Hz, 1H, H-14), 1.70 (dm, *J* = 12.4 Hz, 1H, H-14), 1.96 (dd, *J* = 18.0, 12.0 Hz, 1H, H-20), 2.14–2.26 (m, 4H, H-6, CH_2_CO), 2.38 (m, 1H, H-15), 2.67 (ddd, *J* = 18.0, 3.6, 1.2 Hz, 1H, H-20), 3.63 (s, 3H, CH_3_O), 3.82 (dd, *J* = 12.0, 2.4 Hz, 1H, C*H*_2_OH), 3.89 (dd, *J* = 12.0, 7.2 Hz, 1H, C*H*_2_OH), 4.10 (dd, *J* = 12.4, 4.4 Hz, 1H, H-3), 4.64 (m, 1H, H-5), 6.92 (d, *J* = 7.6 Hz, 1H, H_AR_), 7.00 (d, *J* = 7.6 Hz, 1H, H_AR_), 7.07 (td, *J* = 7.6, 0.8 Hz, 1H, H_AR_), 7.29 (td, *J* = 7.6, 1.2 Hz, 1H, H_AR_), 7.88 (br. s, 1H, NH); ^13^C-NMR (100.6 MHz, CDCl_3_): *δ* = 29.0 (C-15), 29.6 (C-14), 36.7 (C-6), 37.5 (C-20), 39.6 (*C*H_2_CO), 51.7 (CH_3_O), 56.2 (C-7), 60.0 (C-5), 64.5 (C-3), 66.5 (CH_2_OH), 110.7 (CH_AR_), 123.2 (CH_AR_), 123.5 (CH_AR_), 128.9 (CH_AR_), 129.4 (C_AR_), 140.3 (C_AR_), 170.9 (CO), 171.6 (CO), 177.2 (CO); HRMS (ESI) calcd for [C_19_H_22_N_2_O_5_ + H^+^]: 359.1601, found: 359.1601.

### 3.3. Removal of the Hydroxymethyl Substituent

*(1’R,3′S,7′R,8a’R)-3′-Formyl-7′-(methoxycarbonylmethyl)-2,5′-dioxospiro[indoline-3,1′-indolizidine]* (**5**). Dess-Martin periodinane (DMP, 649 mg, 1.53 mmol) and NaHCO_3_ (257 mg, 3.06 mmol) were added at room temperature under an argon atmosphere to a solution of spirooxindole **4** (367 mg, 1.02 mmol) in anhydrous CH_2_Cl_2_ (38 mL). The resulting mixture was stirred at room temperature for 4 h. Saturated aqueous solutions of Na_2_S_2_O_3_ (30 mL) and NaHCO_3_ (30 mL) were added and the mixture was stirred for 30 min. The aqueous layer was extracted with CH_2_Cl_2_. The combined organic extracts were dried, filtered, and concentrated under reduced pressure. Flash chromatography (1:1 hexane-EtOAc to EtOAc) of the resulting residue gave aldehyde **5** (280 mg, 77%) as a white foam: ^1^H-NMR (400 MHz, CDCl_3_, COSY, g-HSQC): *δ* = 0.75 (q, *J* = 12.4 Hz, 1H, H-14), 1.73 (dm, *J* = 12.4 Hz, 1H, H-14), 1.99 (dd, *J* = 17.8, 12.0 Hz, 1H, H-20), 2.14–2.33 (m, 3H, CH_2_CO, H-6), 2.44–2.53 (m, 2H, H-6, H-15), 2.70 (ddd, *J* = 17.8, 5.4, 1.9 Hz, 1H, H-20), 3.63 (s, 3H, CH_3_O), 4.15 (dd, *J* = 11.4, 4.2 Hz, 1H, H-3), 4.88 (t, *J* = 9.6 Hz, 1H, H-5), 6.90 (d, *J* = 7.4 Hz, 1H, H_AR_), 6.97 (d, *J* = 7.8 Hz, 1H, H_AR_), 7.07 (td, *J* = 7.5, 1.0 Hz, 1H, H_AR_), 7.30 (td, *J* = 7.6, 0.4 Hz, 1H, H_AR_), 7.69 (br. s, 1H, NH), 9.73 (d, *J* = 1.7 Hz, 1H, CHO); ^13^C-NMR (100.6 MHz, CDCl_3_): *δ* = 29.5 (C-15), 29.7 (C-14), 34.6 (C-6), 37.2 (C-20), 39.7 (*C*H_2_CO), 51.7 (CH_3_O), 56.7 (C-7), 62.9 (C-5), 65.0 (C-3), 110.8 (CH_AR_), 123.3 (CH_AR_), 123.5 (CH_AR_), 128.8 (CH_AR_), 129.1 (C_AR_), 140.2 (C_AR_), 169.1 (CO), 171.6 (CO), 176.4 (CO), 197.4 (CHO); HRMS (ESI) calcd for [C_19_H_20_N_2_O_5_ + H^+^]: 357.1445, found: 357.1450.

*(1′R,3′S,7′R,8a’R)-7′-(Methoxycarbonylmethyl)-2,5′-dioxo-3′-(phenylselenocarbonyl)spi-ro[indoline-3,1′-indolizidine]* (**7**). First step: 2-Methyl-2-butene (2 M in hexane, 6.4 mL) and *t*-BuOH (25.2 mL) were added at room temperature to a solution of aldehyde **5** (321 mg, 0.9 mmol) in CH_3_CN (8.1 mL). A solution of NaClO_2_ (472 mg, 5.22 mmol) and NaH_2_PO_4_ (733 mg, 5.31 mmol) in distilled H_2_O (8.7 mL) was added at 0 °C to the above mixture, which was stirred at room temperature for 1 h. Then, 0.1 M Na_2_S_2_O_3_ and 2 N HCl solutions were added until pH = 1, and the resulting mixture was extracted with EtOAc, dried over anhydrous MgSO_4_, filtered, and concentrated under reduced pressure to give carboxylic acid **6**, which was used in the next step without purification. Second step: Diphenyl diselenide [(PhSe)_2_, 478 mg, 1.53 mmol] and tri-*n*-butylphosphine (*n*-Bu_3_P, 0.63 mL, 2.52 mmol) were added at room temperature under an argon atmosphere to a solution of crude acid **6** in anhydrous CH_2_Cl_2_ (5.4 mL). The resulting mixture was stirred at reflux for 16 h. Distilled H_2_O was added. The aqueous layer was extracted with CH_2_Cl_2_, and the combined organic extracts were dried over anhydrous MgSO_4_, filtered, and concentrated under reduced pressure. Flash chromatography (hexane to 7:3 hexane-EtOAc) of the resulting residue gave seleno ester **7** (216 mg, 50% overall yield for the two steps) as a yellow foam: [α]^22^_D_ = −4.4 (*c* 1.04, CHCl_3_); IR (film): 3242 (NH), 1726, 1698, 1660, 1635 (CO) cm^−1^; ^1^H-NMR (400 MHz, CDCl_3_, COSY, g-HSQC): *δ* = 0.73 (q, *J* = 12.8 Hz, 1H, H-14), 1.75–1.82 (dm, *J* = 12.8 Hz, 1H, H-14), 1.99 (dd, *J* = 18.0, 12.0 Hz, 1H, H-20), 2.17 (dd, *J* = 15.6, 7.6 Hz, 1H, CH_2_CO), 2.30 (dd, *J* = 15.6, 6.4 Hz, 1H, CH_2_CO), 2.46 (dd, *J* = 12.8, 8.4 Hz, 1H, H-6), 2.46–2.58 (m, 1H, H-15), 2.62 (dd, *J* = 12.8, 8.4 Hz, 1H, H-6), 2.73–2.79 (dm, *J* = 18.0 Hz, 1H, H-20), 3.64 (s, 3H, CH_3_O), 4.37 (dd, *J* = 11.2, 4.4 Hz, H-3), 5.13 (t, *J* = 9.2 Hz, 1H, H-5), 6.84 (d, *J* = 7.6 Hz, 1H, H_AR_), 6.99 (d, *J* = 7.6 Hz, 1H, H H_AR_), 7.05 (td, *J* = 7.6, 1.2 Hz, 1H, H_AR_), 7.29 (td, *J* = 8.0, 1.2 Hz, 1H, H_AR_), 7.36–7.39 (m, 3H, H_AR_), 7.52–7.55 (m, 2H, H_AR_), 8.74 (s, 1H, NH); ^13^C-NMR (100.6 MHz, CDCl_3_): *δ* = 29.3 (C-15), 29.6 (C-14), 37.5 (C-20), 38.0 (C-6), 39.6 (*C*H*_2_*CO), 51.8 (CH_3_O), 57.1 (C-7), 65.6 (C-3), 66.4 (C-5), 110.8 (CH_AR_), 123.3 (CH_AR_), 123.4 (CH_AR_), 124.8 (C_AR_), 128.8–129.4 (4CH_AR_, C_AR_), 136.1 (2CH_AR_), 140.2 (C_AR_), 169.4 (CO), 171.6 (CO), 176.1 (CO), 200.5 (CO); HRMS (ESI) calcd for [C_25_H_24_N_2_O_5_Se + H^+^]: 513.0923, found: 513.0927.

*(1′R,7′R,8a’R)-7′-(Methoxycarbonylmethyl)-2,5′-dioxospiro[indoline-3,1′-indolizidine]* (**8**). Method A: from seleno derivative **7**: Azobisisobutyronitrile (AIBN, 9 mg, 0.05 mmol) was added under an argon atmosphere to a solution of seleno derivative **7** (216 mg, 0.43 mmol) in anhydrous benzene (20 mL). The mixture was heated to reflux, and a solution of tributyltin hydride (TBTH, 180 µL, 0.65 mmol) in anhydrous benzene (4 mL) was added very slowly (over 30 min). The resulting mixture was stirred at reflux for 1 h, and the solvent was evaporated. Flash chromatography (6:4 hexane-EtOAc to 100% EtOAc) of the resulting residue gave spirooxindole **8** (127 mg, 90%). Method B: from aldehyde **5**: Argon was bubbled through anhydrous diglyme (3.2 mL) for 30 min. Chloro(1,5-cyclooctadiene)rhodium(I) dimer (3 mg, 0.006 mmol) and 1,3-bis(diphenylphosphino) propane (dppp, 10 mg, 0.023 mmol) were weighed in corning tubes and introduced into the reaction flask under an argon flow using inert glovebox equipment. Anhydrous diglyme (2.2 mL) was transferred into the reaction flask and the bubbling of argon was continued for 15 min. Aldehyde **5** (80 mg, 0.23 mmol) was dissolved in anhydrous diglyme and transferred into the flask. The mixture was stirred at reflux for 24 h. Distilled H_2_O (2.2 mL) and CH_2_Cl_2_ (2.2 mL) were added, the layers were separated, and the aqueous phase was extracted with CH_2_Cl_2_. The combined organic extracts were washed with brine, dried over anhydrous MgSO_4_, filtered, and concentrated under reduced pressure. Flash chromatography (1:1 to 1:9 hexane-EtOAc) of the resulting residue gave compound **8** (57 mg, 75%) as a white foam and minor amounts of its 5,6-dehydro derivative, which was hydrogenated (10% Pd/C, absolute EtOH) to give additional compound **8** (3 mg, 4%) after flash chromatography: [α]^22^_D_ = + 63.0 (*c* 0.55, CHCl_3_); IR (film): 3194 (NH), 1727, 1619 (CO) cm^−1^; ^1^H-NMR (400 MHz, CDCl_3_, COSY, g-HSQC): *δ* = 0.72 (q, *J* = 12.4 Hz, 1H, H-14), 1.67 (dm, *J* = 12.4 Hz, 1H, H-14), 1.95 (dd, *J* = 17.6, 12.0 Hz, 1H, H-20), 2.09 (dd, *J* = 12.4, 8.4 Hz, 1H, H-6), 2.14 (dd, *J* = 15.6, 7.6 Hz, 1H, CH_2_CO), 2.23 (dd, *J* = 15.6, 6.4 Hz, 1H, CH_2_CO), 2.35 (m, 1H, H-15), 2.52 (q, *J* = 12.4 Hz, 1H, H-6), 2.61 (dd, *J* = 17.6, 4.8 Hz, 1H, H-20), 3.61 (s, 3H, CH_3_O), 3.83 (t, *J* = 11.2 Hz, 1H, H-5), 3.99 (tm, *J* = 11.2 Hz, 1H, H-5), 4.04 (dd, *J* = 11.2, 4.4 Hz, 1H, H-3), 6.91 (d, *J* = 7.2 Hz, 1H, H_AR_), 6.99 (d, *J* = 7.6 Hz, 1H, H_AR_), 7.05 (t, *J* = 7.6 Hz, 1H, H_AR_), 7.28 (t, *J* = 7.6 Hz, 1H, H_AR_), 8.92 (br. s, 1H, NH); ^13^C-NMR (100.6 MHz, CDCl_3_): *δ* = 29.6 (C-15), 29.7 (C-14), 33.3 (C-6), 37.3 (C-20), 39.9 (*C*H_2_CO), 43.9 (C-5), 51.7 (CH_3_O), 56.9 (C-7), 64.3 (C-3), 110.5 (CH_AR_), 123.1 (CH_AR_), 123.7 (CH_AR_), 128.6 (CH_AR_), 129.7 (C_AR_), 140.2 (C_AR_), 168.4 (CO), 171.7 (CO), 177.5 (CO); HRMS (ESI) calcd for [C_18_H_20_N_2_O_4_ + H^+^]: 329.1496, found: 329.1497.

### 3.4. Introduction of the E-Ethylidene Chain

*(1’R,7′S,8a’R)-7′-(2-Hydroxyethyl)-2,5′-dioxospiro[indoline-3,1′-indolizidine]* (**9**): Lithium borohydride (LiBH_4_, 20 mg, 0.9 mmol) was added at 0 °C under an argon atmosphere to a solution of compound **8** (49 mg, 0.15 mmol) in anhydrous THF (5 mL). The resulting mixture was stirred at room temperature for 72 h. The reaction was quenched at 0 °C by distilled H_2_O (5 mL), and the mixture was concentrated under reduced pressure using a rotary evaporator with a dry ice condenser. Flash chromatography (95:5 EtOAc-MeOH) of the resulting residue gave oxindole **9** (34 mg, 76%) as a white foam and minor amounts of indoline **10** (3 mg, 5%). Oxindole **9**: [α]^22^_D_ = + 58,0 (*c* 1.12, MeOH); IR (film): 3100–3600 (NH, OH), 1731, 1714 (CO) cm^−1^; ^1^H-NMR (400 MHz, CDCl_3_, COSY, g-HSQC): *δ* = 0.64 (q, *J* = 12.0 Hz, 1H, H-14), 1.43 (m, 2H, C*H*_2_CH_2_O), 1.64 (dm, *J* = 12.0 Hz, 1H, H-14), 1.90 (dd, *J* = 17.6, 12.0 Hz, 1H, H-20), 2.05 (m, 2H, H-6, H-15), 2.49 (dt, *J* = 10.4 Hz, 1H, H-6), 2.58 (dd, *J* = 17.6, 5.2 Hz, 1H, H-20), 3.60 (t, *J* = 6.4 Hz, 2H, CH_2_O), 3.80 (dd, *J* = 11.6, 9.2 Hz, 1H, H-5), 3.98 (m, 2H, H-5, H-3), 6.91 (d, *J* = 7.6 Hz, 1H, H_AR_), 6.97 (d, *J* = 8.0 Hz, 1H, H_AR_), 7.04 (t, *J* = 7.6 Hz, 1H, H_AR_), 7.26 (td, *J* = 7.6, 1.2 Hz, 1H, H_AR_), 8.83 (br. s, 1H, NH); ^13^C-NMR (100.6 MHz, CDCl_3_): *δ* = 29.5 (C-15), 29.9 (C-14), 33.2 (C-6), 37.9 (C-20), 38.4 (*C*H_2_CH_2_O), 43.8 (C-5), 57.0 (C-7), 59.7 (CH_2_O), 64.6 (C-3), 110.3 (CH_AR_), 123.1 (CH_AR_), 123.8 (CH_AR_), 128.6 (CH_AR_), 129.9 (C_AR_), 140.1 (C_AR_), 169.2 (CO), 177.5 (CO); HRMS (ESI) calcd for [C_17_H_20_N_2_O_3_ + Na^+^]: 323.1366, found: 323.1371. *(1′S,7′S,8a’R)-7′-(2-Hydroxyethyl)-5′-oxospiro[indoline-3,1′-indolizidine]* (**10**): IR (film): 3346 (NH, OH), 1606 (CO) cm^−1^; ^1^H-NMR (400 MHz, CDCl_3_, COSY, g-HSQC): *δ* = 0.69 (q, *J* = 12.4 Hz, 1H, H-14), 1.48 (m, 2H, C*H*_2_CH_2_O), 1.83–1.92 (m, 2H, H-14, H-20), 1.97–2.08 (m, 2H, H-15, H-6), 2.25 (ddd, *J* = 12.8, 8.4, 2.0 Hz, 1H, H-6), 2.54 (ddd, *J* = 17.6, 5.2, 1.6 Hz, 1H, H-20), 3.50–3.66 (m, 2H, H-5, H-3), 3.51 (d, *J* = 9.2 Hz, 1H, H-2), 3.57 (d, *J* = 9.2 Hz, 1H, H-2), 3.65 (t, *J* = 6.4 Hz, 2H, CH_2_C*H*_2_O), 3.83–3.91 (m, 1H, H-5), 6.66 (d, *J* = 7.6 Hz, 1H, H_AR_), 6.71 (t, *J* = 7.6 Hz, 1H, H_AR_), 6.77 (dd, *J* = 7.6, 1.6 Hz, 1H, H_AR_), 7.08 (td, *J* = 7.6, 1.2 Hz, 1H, H_AR_); ^13^C-NMR (100.6 MHz, CDCl_3_): *δ* = 29.7 (C-15), 30.3 (C-14), 35.9 (C-6), 38.2 (C-20), 38.7 (*C*H_2_CH_2_O), 43.5 (C-5), 54.9 (C-2), 55.4 (C-7), 60.0 (CH_2_*C*H_2_O), 66.1 (C-3), 110.1 (CH_AR_), 119.5 (CH_AR_), 124.0 (CH_AR_), 128.5 (CH_AR_), 131.0 (C_AR_), 151.2 (C_AR_), 169.2 (CO); HRMS (ESI) calcd for [C_17_H_22_N_2_O_2_ + H^+^]: 287.1754, found: 287.1758.

*(1′R,7′S,8a’R)-7′-{2-[(tert-Butyldimethylsilyl)oxy]ethyl}-2,5′-dioxospiro[indoline-3,1′-indolizidine]* (**11**). *tert*-Butyldimethylsilyl chloride (TBDMSCl, 27 mg, 0.18 mmol) and imidazole (25 mg, 0.36 mmol) were added at 0 °C under an argon atmosphere to a solution of alcohol **9** (27 mg, 0.09 mmol) in anhydrous DMF (1 mL). The resulting mixture was stirred at room temperature for 16 h. Brine was added, and the mixture was extracted with EtOAc. The combined organic extracts were dried over anhydrous MgSO_4_, filtered, and concentrated under reduced pressure. Flash chromatography (7:3 hexane-EtOAc to 1:1 hexane-EtOAc) of the resulting residue gave silyl derivative **11** (30 mg, 80%) as a white foam: [α]^22^_D_ = + 37.09 (*c* 0.23, CHCl_3_); IR (film): 3181 (NH), 1725, 1620 (CO) cm^−1^; ^1^H-NMR (400 MHz, CDCl_3_, COSY, g-HSQC): *δ* = −0.01 (s, 3H, CH_3_Si), 0.00 (s, 3H, CH_3_Si), 0.65 (qd, *J* = 12.4, 3.6 Hz, 1H, H-14), 0.83 [s, 9H, C(CH_3_)_3_], 1.39 (m, 2H, C*H*_2_CH_2_O), 1.65 (dm, *J* = 12.4 Hz, 1H, H-14), 1.89 (dd, *J* = 16.4, 11.6 Hz, 1H, H-20), 2.03–2.09 (m, 2H, H-15, H-6), 2.50–2.58 (m, 2H, H-6, H-20), 3.55 (m, 2H, CH_2_O), 3.81 (t, *J* = 12.4 Hz, 1H, H-5), 3.95–4.03 (m, 2H, H-5, H-3), 6.91–6.95 (m, 2H, H_AR_), 7.03–7.07 (m, 1H, H_AR_), 7.27 (m, 1H, H_AR_), 7.68 (br. s, 1H, NH); ^13^C-NMR (100.6 MHz, CDCl_3_): *δ* = −5.5 (2CH_3_Si), 18.2 [*C*(CH_3_)_3_], 25.8 [C(*C*H_3_)_3_], 29.7 (C-15), 30.1 (C-14), 33.3 (C-6), 37.9 (C-20), 38.5 (*C*H_2_CH_2_O), 43.8 (C-5), 57.0 (C-7), 60.1 (CH_2_O), 64.8 (C-3), 110.2 (CH_AR_), 123.1 (CH_AR_), 123.9 (CH_AR_), 128.5 (CH_AR_), 129.9 (C_AR_), 140.0 (C_AR_), 169.2 (CO), 177.5 (CO); HRMS (ESI) calcd for [C_23_H_34_N_2_O_3_Si + H^+^]: 415.2411, found: 415.2419.

*(1′R,7′S,8a’R)-7′-{2-[(tert-Butyldimethylsilyl)oxy]ethyl}-1-(methoxymethyl)-2,5′-dioxospiro[indoline-3,1′-indolizidine]* (**12**). A solution of compound **11** (386 mg, 0.93 mmol) in anhydrous THF (2.5 mL) was transferred at 0 °C under an argon atmosphere to a suspension of NaH (95%, 36 mg, 1.4 mmol) in anhydrous DMF (2.5 mL). The resulting mixture was stirred at 0 °C for 30 min. Methoxymethyl chloride (MOMCl, 0.12 mL, 1.4 mmol) was added, and the resulting mixture was stirred at room temperature for 1.5 h. The mixture was cooled to 0 °C and saturated aqueous NaHCO_3_ (11.2 mL) was added. The mixture was extracted with EtOAc and the combined organic extracts were dried over anhydrous MgSO_4_, filtered, and concentrated under reduced pressure. Flash chromatography (hexane to 7:3 hexane-EtOAc) of the resulting residue gave the *N*-MOM derivative **12** (283 mg, 67%) as a white foam: [α]^22^_D_ = + 50.6 (*c* 2.26, CHCl_3_); IR (film): 1725, 1651 (CO) cm^−1^; ^1^H-NMR (400 MHz, CDCl_3_, COSY, g-HSQC): *δ* = −0.02 (s, 3H, CH_3_Si), −0.03 (s, 3H, CH_3_Si), 0.64 (q, *J* = 12.4 Hz, 1H, H-14), 0.82 [s, 9H, C(CH_3_)_3_], 1.37 (dd, *J* = 12.8, 6.4 Hz, 2H, C*H*_2_CH_2_O), 1.55 (dm, *J* = 12.4 Hz, 1H, H-14), 1.89 (dd, *J* = 17.6, 12.0 Hz, 1H, H-20), 2.00–2.07 (m, 2H, H-15, H-6), 2.50–2.57 (m, 2H, H-20, H-6), 3.32 (s, 3H, CH_3_O), 3.54 (t, *J* = 6.0 Hz, 2H, CH_2_C*H*_2_O), 3.81 (t, *J* = 11.6 Hz, 1H, H-5), 3.96–4.03 (m, 2H, H-5, H-3), 5.15 (s, 2H, NCH_2_O), 6.94 (d, *J* = 7.6 Hz, 1H, H_AR_), 7.08–7.11 (m, 2H, H_AR_), 7.32 (t, *J* = 7.6 Hz, 1H, H_AR_); ^13^C-NMR (100.6 MHz, CDCl_3_): *δ* = −5.5 (2CH_3_Si), 18.2 [*C*(CH_3_)_3_], 25.8 [C(*C*H_3_)_3_], 29.6 (C-15), 30.2 (C-14), 33.7 (C-6), 37.8 (C-20), 38.6 (*C*H_2_CH_2_O), 43.8 (C-5), 56.3, 56.9 (C-7, CH_3_O), 60.0 (CH_2_*C*H_2_O), 64.8 (C-3), 71.5 (NCH_2_O), 110.0 (CH_AR_), 123.6 (CH_AR_), 123.7 (CH_AR_), 128.6 (CH_AR_), 129.1 (C_AR_), 141.2 (C_AR_), 169.1 (CO), 176.3 (CO); HRMS (ESI) calcd for [C_25_H_38_N_2_O_4_Si + H^+^]: 459.2674, found: 459.2675.

*(1′R,7′S,8a’R)-7′-{2-[(tert-Butyldimethylsilyl)oxy]ethyl}-1-(tert-butoxycarbonyl)-2,5′-dioxospiro[indoline-3,1′-indolizidine]* (**13**). NaH (95%, 25 mg, 0.64 mmol) and (Boc)_2_O (70 mg, 0.32 mmol) were added at 0 °C under an argon atmosphere to a solution of compound **11** (33 mg, 0.08 mmol) in anhydrous DMF (1 mL). The mixture was stirred at room temperature for 16 h. The reaction was quenched by the addition of a few drops of distilled H_2_O, and the resulting residue was dried over anhydrous MgSO_4_, filtered, and concentrated under reduced pressure. Flash chromatography (hexane to 4:1 EtOAc-MeOH) gave *N*-Boc derivative **13** (26 mg, 64%): [α]^22^_D_ = + 56.7 (*c* 0.92, CHCl_3_); IR (film): 1794, 1766, 1733 (CO) cm^−1^; ^1^H-NMR (400 MHz, CDCl_3_, COSY, g-HSQC): *δ* = −0.28 (s, 3H, CH_3_Si), −0.21 (s, 3H, CH_3_Si), 0.60 (q, *J* = 12.0 Hz, 1H, H-14), 0.82 [s, 9H, SiC(CH_3_)_3_], 1.37–1.45 (m, 2H, C*H*_2_CH_2_O), 1.65 [s, 9H, C(CH_3_)_3_], 1.65 (masked, 1H, H-14), 1.88 (dd, *J* = 17.6, 12.0 Hz, 1H, H-20), 2.01 (m, 1H, H-15), 2.08 (dd, *J* = 12.8, 6.8 Hz, 1H, H-6), 2.52 (m, 2H, H-6, H-20), 3.50–3.57 (m, 2H, CH_2_C*H*_2_O), 3.79 (t, *J* = 12.0 Hz, 1H, H-5), 3.94–4.03 (m, 2H, H-3, H-5), 6.90 (d, *J* = 6.8 Hz, 1H, H_AR_), 7.16 (t, *J* = 7.6 Hz, 1H, H_AR_), 7.34 (td, *J* = 7.2, 1.2 Hz, 1H, H_AR_), 7.90 (d, *J* = 8.0 Hz, 1H, H_AR_); ^13^C-NMR (100.6 MHz, CDCl_3_): *δ* = −5.5 (2CH_3_Si), 18.1 [Si*C*(CH_3_)_3_], 25.8 [SiC(*C*H_3_)_3_], 28.1 [C(*C*H_3_)_3_], 29.9 (C-15), 30.2 (C-14), 34.5 (C-6), 37.8 (C-20), 38.5 (*C*H_2_CH_2_O), 43.7 (C-5), 56.8 (C-7), 60.2 (CH_2_*C*H_2_O), 65.5 (C-3), 84.9 [*C*(CH_3_)_3_], 115.3 (CH_AR_), 123.3 (CH_AR_), 125.1 (CH_AR_), 128.5 (C_AR_), 128.7 (CH_AR_), 138.9 (C_AR_), 148.9 (CO), 169.1 (CO), 174.6 (CO); HRMS (ESI) calcd for [C_28_H_42_N_2_O_5_Si + H^+^]: 515.2936, found: 515.2941.

*(1′R,6′R,7′S,8a’R)-6′-Acetyl-7′-{2-[(tert-butyldimethylsilyl)oxy]ethyl}-1-(methoxymtehyl)-2,5′-dioxospiro[indoline-3,1′-indolizidine*] (**14**). Lithium diisopropylamide (LDA, 0.26 mL of a 2.0 M solution in THF/heptane/ethylbenzene, 0.42 mmol) was added at −78 °C under an argon atmosphere to a solution of spiro compound **12** (62 mg, 0.14 mmol) in anhydrous THF (0.7 mL), and the mixture was stirred at −78 °C for 1 h. Methyl acetate (0.05 mL, 0.56 mmol) was added at −78 °C, and the resulting mixture was stirred at room temperature for 4 h. Saturated aqueous NH_4_Cl was added, and the mixture was extracted with CH_2_Cl_2_. The combined organic extracts were dried over anhydrous MgSO_4_, filtered, and concentrated under reduced pressure. Flash chromatography (hexane to 7:3 hexane-EtOAc) of the resulting residue gave starting material **12** (18 mg, 29%) and acetyl derivative **14** (35 mg, 52%) as a white foam: [α]^22^_D_ = + 26.2 (c 0.82, CHCl_3_); IR (film): 1723, 1642, 1613 (CO) cm^−1^; ^1^H-NMR (400 MHz, CDCl_3_, COSY, g-HSQC): δ = −0.07 (s, 3H, CH_3_Si), −0.06 (s, 3H, CH_3_Si), 0.70–0.84 (m, 1H, H-14), 0.79 [s, 9H, C(CH_3_)_3_], 1.19–1.25 (m, 1H, CH_2_CH_2_O), 1.41–1.49 (m, 1H, CH_2_CH_2_O), 1.64–1.70 (m, 1H, H-14), 2.05 (dd, J = 12.8, 2.8 Hz, 1H, H-6), 2.37 (s, 3H, COCH_3_), 2.40–2.46 (m, 1H, H-15), 2.55 (dt, J = 12.8, 10.8 Hz, 1H, H-6), 3.21 (d, J = 10.8 Hz, 1H, H-20), 3.32 (s, 3H, CH_3_O), 3.47 (t, J = 6.4 Hz, 2H, CH_2_CH_2_O), 3.81 (dd, J = 12.8, 10.8 Hz, 1H, H-5), 3.96–4.02 (m, 1H, H-5), 4.09 (dd, J = 11.6, 4.0 Hz, 1H, H-3), 5.13 (d, J = 10.8 Hz, 1H, NCH_2_O), 5.15 (d, J = 10.8 Hz, 1H, NCH_2_O), 6.92 (d, J = 7.6 Hz, 1H, H_AR_), 7.08–7.12 (m, 2H, H_AR_), 7.33 (td, J = 9.2, 1.2 Hz, 1H, H_AR_); ^13^C-NMR (100.6 MHz, CDCl_3_): δ = −5.6 (CH_3_Si), −5.5 (CH_3_Si), 18.1 [C(CH_3_)_3_], 25.8 [C(CH_3_)_3_], 28.8 (C-14), 31.1 (CH_3_CO), 32.7 (C-15), 33.7 (C-6), 36.6 (CH_2_CH_2_O), 44.3 (C-5), 56.3 (CH_3_O), 56.9 (C-7), 60.2 (CH_2_CH_2_O), 61.7 (C-20), 64.3 (C-3), 71.5 (NCH_2_O), 110.1 (CH_AR_), 123.6 (CH_AR_), 123.7 (CH_AR_), 128.7 (C_AR_), 128.8 (CH_AR_), 141.2 (C_AR_), 165.8 (CO), 176.0 (CO), 205.2 (CO); HRMS (ESI) calcd for [C_27_H_40_N_2_O_5_Si + H^+^]: 501.2779, found: 501.2791.

*(1′R,6′R,7′S,8a’R)-7′-{2-[(tert-Butyldimethylsilyl)oxy]ethyl}-6′-(1R- and 1S-hydroxyethyl)-1-(methoxymethyl)-2,5′-dioxospiro[indoline-3,1′-indolizidine]* (**15a** and **15b**). NaBH_4_ (10 mg, 0.24 mmol) was added at −10 °C under an argon atmosphere to a solution of ketone **14** (60 mg, 0.12 mmol) in anhydrous MeOH (2 mL). The resulting mixture was stirred at −10 °C for 1 h. Saturated aqueous NaHCO_3_ (1.3 mL) and CH_2_Cl_2_ were added, and the mixture was stirred for 5 min. The organic solvent was evaporated, and the resulting aqueous mixture was extracted with CH_2_Cl_2_. The combined organic extracts were dried over anhydrous MgSO_4_, filtered, and concentrated under reduced pressure. Flash chromatography (7:3 hexane-EtOAc to 1:1 hexane-EtOAc) of the resulting residue gave alcohols **15a** (28 mg, 46%) and **15b** (27 mg, 46%) as white foams. **15a**: [α]^22^_D_ = + 30.3 (c 1.16, CHCl_3_); IR (film): 3427 (OH), 1725, 1614 (CO) cm^−1^; ^1^H-NMR (400 MHz, CDCl_3_, COSY, g-HSQC): δ = −0.05 (s, 3H, CH_3_Si), −0.04 (s, 3H, CH_3_Si), 0.65–0.81 (m, 1H, H-14), 0.81 [s, 9H, C(CH_3_)_3_], 1.24–1.32 (m, 1H, CH_2_CH_2_O), 1.39 (d, J = 6.4 Hz, 3H, CH_3_CHOH), 1.58 (dt, J = 13.2, 3.9 Hz, 1H, H-14), 1.76–1.85 (m, 1H, CH_2_CH_2_O), 1.92–1.95 (m, 1H, H-15), 2.00–2.05 (m, 1H, H-6), 2.11 (dd, J = 9.2, 4.8 Hz, 1H, H-20), 2.54 (td, J = 12.4, 9.6 Hz, 1H, H-6), 3.32 (s, 3H, CH_3_O), 3.45–3.56 (m, 2H, CH_2_CH_2_O), 3.81 (dd, J = 12.4, 9.6 Hz, 1H, H-5), 3.96–4.03 (m, 3H, H-3, H-5, CHOH), 5.15 (s, 2H, NCH_2_O), 6.96 (dm, J = 8.0 Hz, 1H, H_AR_), 7.08–7.12 (m, 2H, H_AR_), 7.33 (td, J = 8.0, 1.2 Hz, 1H, H_AR_); ^13^C-NMR (100.6 MHz, CDCl_3_): δ = −5.5 (2CH_3_Si), 18.1 [C(CH_3_)_3_], 22.1 (CH_3_CHOH), 25.8 [C(CH_3_)_3_], 30.4 (C-14), 32.4 (C-15), 33.9 (C-6), 38.0 (CH_2_CH_2_O), 44.7 (C-5), 53.3 (C-20), 56.3 (CH_3_O), 57.2 (C-7), 60.7 (CH_2_CH_2_O), 63.6 (C-3), 69.7 (CHOH), 71.5 (NCH_2_O), 110.1 (CH_AR_), 123.6 (CH_AR_), 123.8 (CH_AR_), 128.7 (CH_AR_, C_AR_), 141.2 (C_AR_), 171.2 (CO), 176.2 (CO); HRMS (ESI) calcd for [C_27_H_42_N_2_O_5_Si + H^+^]: 503.2936, found: 503.2937. **15b**: [α]^22^_D_ = + 14.0 (c 1.13, CHCl_3_); IR (film): 3418 (OH), 1725, 1614 (CO) cm^−1^; ^1^H-NMR (400 MHz, CDCl_3_, COSY, g-HSQC): δ = −0.07 (s, 3H, CH_3_Si), −0.05 (s, 3H, CH_3_Si), 0.70–0.84 (m, 1H, H-14), 0.80 [s, 9H, C(CH_3_)_3_], 1.18–1.20 (m, 1H, CH_2_CH_2_O), 1.19 (d, J = 6.4 Hz, 3H, CH_3_CHOH), 1.61–1.75 (m, 3H, H-14, H-15, CH_2_CH_2_O), 2.06 (dd, J = 12.8, 6.4 Hz, 1H, H-6), 2.34 (dd, J = 10.8, 3.6 Hz, 1H, H-20), 2.55 (tm, J = 12.8 Hz, 1H, H-6), 3.32 (s, 3H, CH_3_O), 3.44–3.54 (m, 2H, CH_2_CH_2_O), 3.84 (dd, J = 12.8, 10.8 Hz, 1H, H-5), 3.96–4.01 (m, 3H, H-3, H-5, CHOH), 5.15 (s, 2H, NCH_2_O), 6.94 (d, J = 7.2 Hz, 1H, H_AR_), 7.09–7.13 (m, 2H, H_AR_), 7.34 (t, J = 7.6 Hz, 1H, H_AR_); ^13^C-NMR (100.6 MHz, CDCl_3_): δ = −5.5 (2CH_3_Si), 18.0 [C(CH_3_)_3_], 18.6 (CH_3_CHOH), 25.8 [C(CH_3_)_3_], 29.3 (C-14), 32.2 (C-15), 33.8 (C-6), 36.3 (CH_2_CH_2_O), 44.1 (C-5), 52.3 (C-20), 56.3 (C-7), 57.0 (CH_3_O), 59.9 (CH_2_CH_2_O), 64.1 (C-3), 67.1 (CHOH), 71.5 (NCH_2_O), 110.2 (CH_AR_), 123.7 (CH_AR_), 123.8 (CH_AR_), 128.7 (C_AR_), 128.8 (CH_AR_), 141.2 (C_AR_), 172.2 (CO), 176.1 (CO); HRMS (ESI) calcd for [C_27_H_42_N_2_O_5_Si + H^+^]: 503.2936, found: 503.2933.

*(1′R,6′E,7′S,8a’R)-7′-{2-[(tert-Butyldimethylsilyl)oxy]ethyl}-6′-ethylidene-1-(methoxymethyl)-2,5′-dioxospiro[indoline-3,1′-indolizidine]* (**16**). From alcohol **15a**: *N*,*N*′-Dicyclohexylcarbodiimide (DCC, 54 mg, 0.26 mmol) and copper(I) chloride (CuCl, 52 mg, 0.52 mmol) were added under an argon atmosphere to a solution of alcohol **15a** (26 mg, 0.05 mmol) in anhydrous toluene (1.6 mL), and the resulting mixture was stirred at reflux for 5 h. The suspension was filtered through Celite^®^, and the residue was washed with CH_3_CN. The resulting filtrate was kept in the freezer overnight and filtered again through Celite^®^, washing with minimal amounts of cold CH_3_CN. The organic filtrate was concentrated under reduced pressure. Flash chromatography (1:9 hexane-EtOAc) of the resulting residue gave the E-ethylidene derivative **16** (22 mg, 91%). From alcohol **15b**: First step: Et_3_N (18 μL, 0.13 mmol) and mesyl chloride (MsCl, 9 μL, 0.11 mmol) were added at 0 °C under an argon atmosphere to a solution of alcohol **15b** (21 mg, 0.04 mmol) in anhydrous CH_2_Cl_2_ (0.6 mL). The resulting mixture was stirred at 0 °C for 4 h. Saturated aqueous NH_4_Cl (1.2 mL) was added, and the mixture was extracted with CH_2_Cl_2_. The combined organic extracts were dried over anhydrous MgSO_4_, filtered, and concentrated under reduced pressure to give the corresponding mesylate, which was used in the next step without further purification. Second step: Diazabicycloundecene (DBU, 27 μL, 0.18 mmol) was added under an argon atmosphere to a solution of the above mesylate in anhydrous THF (0.6 mL), and the resulting mixture was stirred at reflux overnight. Distilled H_2_O was added, and the mixture was extracted with EtOAc. The combined organic extracts were dried over anhydrous MgSO_4_, filtered, and concentrated under reduced pressure. Flash chromatography (hexane to 7:3 hexane-EtOAc) of the residue gave the Z isomer of compound **16** (4 mg, 20%) and the E-ethylidene derivative **16** (12 mg, 60%) as white foams. **16**: [α]^22^_D_ = −2.9 (c 0.76, CHCl_3_); IR (film): 1725, 1613 (CO) cm^−1^; ^1^H-NMR (400 MHz, CDCl_3_, COSY, g-HSQC): δ = −0.1 (s, 3H, CH_3_Si), −0.07 (s, 3H, CH_3_Si), 0.78 [s, 9H, C(CH_3_)_3_], 0.94–1.05 (q, J = 12.0 Hz, 1H, H-14), 1.29–1.36 (m, 1H, CH_2_CH_2_O), 1.65–1.70 (m, 1H, H-14), 1.77–1.85 (m, 4H, CH_2_CH_2_O, =CHCH_3_), 1.87–1.95 (dm, J = 12.4 Hz, 1H, H-6), 2.51 (m, 1H, H-6), 2.87–2.96 (m, 1H, H-15), 3.32 (s, 3H, CH_3_O), 3.36–3.52 (m, 2H, CH_2_CH_2_O), 3.88–4.30 (m, 3H, H-5, H-3), 5.14 (d, J = 10.8 Hz, 1H, NCH_2_O), 5.17 (d, J = 10.8 Hz, 1H, NCH_2_O), 6.67 (m, 1H, =CHCH_3_), 7.09–7.16 (m, 3H, H_AR_), 7.33 (td, J = 7.2, 1.6 Hz, H_AR_); ^13^C-NMR (100.6 MHz, CDCl_3_): δ = −5.5 (2CH_3_Si), 14.4 (=CHCH_3_), 18.1 [C(CH_3_)_3_], 25.8 [C(CH_3_)_3_], 30.8 (C-15), 31.1 (C-14), 34.8 (C-6), 39.4 (CH_2_CH_2_O), 44.6 (C-5), 56.3 (CH_3_O), 57.4 (C-7), 60.4 (CH_2_CH_2_O), 61.6 (C-3), 71.5 (NCH_2_O), 110.1 (CH_AR_), 123.5 (CH_AR_), 124.3 (CH_AR_), 128.7 (CH_AR_), 128.8 (C_AR_), 133.5 (=CHCH_3_), 135.7 (C-20), 141.2 (C_AR_), 167.6 (CO), 176.6 (CO); HRMS (ESI) calcd for [C_27_H_40_N_2_O_4_Si + H^+^]: 485.2830, found: 485.2825. Z-isomer of **16**: IR (film): 1726, 1614 (CO) cm^−1^; ^1^H-NMR (400 MHz, CDCl_3_, COSY, g-HSQC, selected resonances): δ = −0.05 (s, 3H, CH_3_Si), −0.04 (s, 3H, CH_3_Si), 0.81 [s, 9H, C(CH_3_)_3_], 1.54–1.64 (m, 1H, H-14), 1.83 (m, 1H, CH_2_CH_2_O), 2.04 (ddd, J = 12.4, 7.6, 1.6 Hz, 1H, H-6), 2.20 (dd, J = 7.2, 2.0 Hz, 3H, =CHCH_3_), 2.54–2.60 (m, 1H, H-6), 3.32 (s, 3H, CH_3_O), 3.48–3.60 (m, 2H, CH_2_CH_2_O), 3.83–3.89 (m, 1H, H-5), 4.02–4.08 (m, 2H, H-3, H-5), 5.15 (s, 2H, NCH_2_O), 5.94 (qd, J = 7.2, 2.0 Hz, 1H, =CHCH_3_), 6.99 (dd, J = 8.0, 1.2 Hz, 1H, H_AR_), 7.07–7.11 (m, 2H, H_AR_), 7.31 (td, J = 7.6, 1.2 Hz, 1H, H_AR_); ^13^C-NMR (100.6 MHz, CDCl_3_, selected resonances): δ = −5.5 (CH_3_Si), −5.4 (CH_3_Si), 15.8 (=CHCH_3_), 18.1 [C(CH_3_)_3_], 25.8 [C(CH_3_)_3_], 29.8 (C-14), 34.1 (C-6), 36.2 (CH_2_CH_2_O), 44.0 (C-5), 56.3 (CH_3_O), 57.0 (C-7), 60.3 (CH_2_CH_2_O), 63.9 (C-3), 71.4 (NCH_2_O), 109.9 (CH_AR_), 123.6 (CH_AR_), 123.9 (CH_AR_), 128.6 (CH_AR_), 132.1 (C_AR_), 134.2 (=CHCH_3_), 141.2 (C_AR_), 165.5 (CO), 176.4 (CO); HRMS (ESI) calcd for [C_27_H_40_N_2_O_4_Si + H^+^]: 485.2830, found: 485.2825.

*(1′R,6′E,7′S,8a’R)-6′-Ethylidene-7′-(2-hydroxyethyl)-2,5′-dioxospiro[indoline-3,1′-indolizidine]* (**17**). TMSCl (30 μL, 0.23 mmol) and sodium iodide (35 mg, 0.23 mmol) were added at 0 °C under an argon atmosphere to a solution of compound **16** (25 mg, 0.05 mg) in anhydrous CH_3_CN (0.9 mL). The resulting mixture was stirred at 0 °C for 2 h. Saturated aqueous NaHCO_3_ was added and the mixture was extracted with EtOAc. The combined organic extracts were washed with brine, dried over anhydrous MgSO_4_, filtered, and concentrated under reduced pressure. The resulting residue was taken in MeOH (4.7 mL), Et_3_N (22 μL, 0.15 mmol) was added to the solution, and the mixture was stirred at 55 °C for 1 h. Saturated aqueous NH_4_Cl (1.7 mL) was added, the organic solvent was evaporated, and the aqueous mixture was extracted with EtOAc. The combined organic extracts were washed with brine, dried over anhydrous MgSO_4_, filtered, and concentrated under reduced pressure. Flash chromatography (hexane to 1:1 hexane-EtOAc) of the residue gave compound **17** (11 mg, 68%) as a white foam: [α]^22^_D_ = −4.3 (*c* 0.44, CHCl_3_); IR (film): 3500–3000 (NH, OH), 1721, 1651 (CO) cm^−1^; ^1^H-NMR (400 MHz, CDCl_3_, COSY, g-HSQC): *δ* = 1.06 (q, *J* = 12.0 Hz, 1H, H-14), 1.39–1.48 (m, 1H, C*H*_2_CH_2_O), 1.70–1.76 (m, 1H, H-14), 1.80 (dd, *J* = 7.2, 1.2 Hz, 1H, =CHC*H*_3_), 1.83–1.91 (m, 1H, C*H*_2_CH_2_O), 2.01–2.06 (m, 1H, H-6), 2.46–2.54 (m, 1H, H-6), 2.93–3.00 (m, 1H, H-15), 3.43–3.49 (m, 1H, CH_2_C*H*_2_O), 3.53–3.59 (m, 1H, CH_2_C*H*_2_O), 3.88–4.02 (m, 3H, H-5, H-3), 6.71 (qd, *J* = 7.2, 2.0 Hz, 1H, =C*H*CH_3_), 6.97 (d, *J* = 7.6 Hz, 1H, H_AR_), 7.05–7.13 (m, 2H, H_AR_), 7.26–7.30 (m, 1H, H_AR_), 8.01 (br. s, 1H, NH); ^13^C-NMR (100.6 MHz, CDCl_3_): *δ* = 14.5 (=CH*C*H_3_), 30.5 (C-14), 30.9 (C-15), 34.4 (C-6), 38.5 (*C*H_2_CH_2_O), 44.8 (C-5), 57.4 (C-7), 59.9 (CH_2_*C*H_2_O), 61.5 (C-3), 110.5 (CH_AR_), 122.9 (CH_AR_), 124.4 (CH_AR_), 128.6 (CH_AR_), 129.7 (C_AR_), 133.6 (=*C*HCH_3_), 134.7 (C-20), 140.2 (C_AR_), 167.1 (CO), 177.7 (CO); HRMS (ESI) calcd for [C_19_H_22_N_2_O_3_ + H^+^]: 327.1703, found: 327.1702.

## Data Availability

The data presented in this study are available in the article and the Appendix A.

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
