# Peer review of "Studies on the Enantioselective Synthesis of E-Ethylidene-bearing Spiro[indolizidine-1,3′-oxindole] Alkaloids"

_molecules, 2021, doi:10.3390/molecules26020428_

Round 1

Reviewer 1 Report

The manuscript by Amat et al. describes studies on stereocontrolled synthesis of (1’R,6’E,7’S,8a’R)-6’-ethylidene-7’-(2-hydroxyethyl)-2,5’-dioxospiro[indoline-3,1’-indolizidine], an alkaloid-type derivative of spiro[indolizidine-1,3’-oxindole having cis H-3/H-15, E-ethylidene arrangement of substituents. This type of compounds attracts interest of many groups of chemists, and the results presented are continuation of previous studies of Amat et al.

The manuscript is a piece of excellent chemistry and stereochemistry exploited for preparation of a multi-functional product, and is worth to be published in Molecules. However, because preparation of one only compound is presented, the title ‘Studies on the Enantioselective Synthesis of E-Ethylidene-Bearing Spiro[indolizidine-1,3’-oxindole] Alkaloids’ is misleading – it suggests a number of examples of products, and a general procedure, which is not the case. Thus, it should be changed.
Two other, minor corrections suggested: 1. Avoid using compound numbers in the Abstract. 2. Compound numbers in the Experimental Section should be in bold.

Author Response

Point 1: “The title ‘Studies on the Enantioselective Synthesis of E-Ethylidene-Bearing Spiro[indolizidine-1,3’-oxindole] Alkaloids’ is misleading – it suggests a number of examples of products, and a general procedure, which is not the case. Thus, it should be changed.” We have maintained the title as it describes precisely the objective of the work.

Point 2: “Avoid using compound numbers in the Abstract”. We have modified the Abstract. In the new version of the Abstract we are not using compound numbers.

Point 3: “Compound numbers in the Experimental Section should be in bold”. We have corrected this item.

Reviewer 2 Report

This paper described the stereoselective synthesis of spiroindoles with cisH-3/H-15 conformation and alkenes in the E configuration. These synthetic routes are very interesting and will provide an efficient method for the synthesis of natural products with similar skeletons.

It is judged to be acceptable for publication with the following addition of mechanistic comments and corrections.

(1) Describe the dehydration of compound 15a in Scheme 5 using DCC/CuCl, including the mechanism.

(2) Anti-elimination from compound 15b of Scheme 5 seems to produce Z isomer. This mechanism should also be reconsidered.

(3) The compound numbers in the experimental section are all plain. Change to Bold.

Author Response

Point 1: “Describe the dehydration of compound 15a in Scheme 5 using DCC/CuCl, including the mechanism”. We have modified the text of the manuscript, including a short comment on the mechanism. We have added new references about the procedure.

Point 2: “Anti-elimination from compound 15b of Scheme 5 seems to produce Z isomer. This mechanism should also be reconsidered”. We have slightly modified the text of the manuscript and corrected an involuntary error in Scheme 6: alcohol 15a has an R configuration and alcohol 15b has an S configuration. We have also corrected this error in the experimental part and Supporting Information.

Point 3: “The compound numbers in the experimental section are all plain. Change to Bold”. We have corrected this item.

Reviewer 3 Report

Amat and co-workers described the synthesis of Spiroindolizidine-1,3’-oxindole structures, characterized by a cis H-3/H-15 stereochemistry, with a methodology based on their previously reported findings (OrgLett 2017,4050). The work and supporting information are well written and of good scientific quality. Although the synthetic methods reported within the paper were somehow previously described, they should raise the interest of the readership of such a journal. However, i would suggest to better explain what is described within lines 48-62, maybe with a scheme.  What are the synthetic approaches known for the synthesis of the trans H-3/H-15 analogues? I think that a scheme could help an inexperienced reader to appreciate the different synthetic pathways. For these reasons I would recommend the acceptance of this paper after this very minor revision.

Author Response

Point 1: “I would suggest to better explain what is described within lines 48-62, maybe with a scheme.  What are the synthetic approaches known for the synthesis of the trans H-3/H-15 analogues? I think that a scheme could help an inexperienced reader to appreciate the different synthetic pathways”. To clarify the text of lines 48-62 we have added a new scheme (Scheme 2) outlining the different stereochemical behavior of Corynanthe-type indolo[2,3-a]quinolizidine amines and amides in the oxidative rearrangement to oxindole derivatives.

Reviewer 4 Report

Please see attached review.

Author Response

No changes are required.